# Impact of Grip Strength in Patients with Unresectable Hepatocellular Carcinoma Treated with Lenvatinib

**DOI:** 10.3390/cancers12082146

**Published:** 2020-08-03

**Authors:** Kei Endo, Hidekatsu Kuroda, Jo Kanazawa, Takuro Sato, Yudai Fujiwara, Tamami Abe, Hiroki Sato, Youhei Kooka, Takayoshi Oikawa, Kei Sawara, Yasuhiro Takikawa

**Affiliations:** Division of Hepatology, Department of Internal Medicine, Iwate Medical University School of Medicine, Iwate 028-3695, Japan; hikuro@iwate-med.ac.jp (H.K.); jknzw5@gmail.com (J.K.); taku_tonton@yahoo.co.jp (T.S.); kenyon1266@yahoo.co.jp (Y.F.); mokosantokosan@yahoo.co.jp (T.A.); hsato@iwate-med.ac.jp (H.S.); y.kooka0416@gmail.com (Y.K.); tyoikawa@iwate-med.ac.jp (T.O.); ikeraw@gmail.com (K.S.); ytakikaw@iwate-med.ac.jp (Y.T.)

**Keywords:** sarcopenia, grip strength, skeletal muscle index, hepatocellular carcinoma, lenvatinib, modified albumin-bilirubin grade

## Abstract

Although sarcopenia is characterized by a loss of muscle strength and skeletal muscle mass, few studies have evaluated the effect of muscle strength on hepatocellular carcinoma (HCC) patients. We evaluated the impact of sarcopenia-related factors (grip strength (GS) and the skeletal muscle index (SMI)) on the survival among lenvatinib-treated unresectable HCC (u-HCC) patients. This single-center cohort study was conducted at a university hospital. The study population included 63 lenvatinib-treated u-HCC patients managed between April 2018 and April 2020. A decreased GS and decreased SMI were found in 21 (33.3%) and 22 (34.9%) patients, respectively. The overall survival (OS) of the normal GS group was significantly higher than that of the decreased GS group, while that of the normal and decreased SMI groups did not differ markedly. There were no significant differences in the progression-free survival between the normal GS and decreased GS groups or the normal SMI and decreased SMI groups. A multivariate Cox proportional hazards model showed that modified albumin-bilirubin-grade (mALBI) 2b (hazard ratio (HR) 4.39) and a decreased GS (HR 3.55) were independently associated with an increased risk of poor prognosis. In addition to the hepatic functional reserve, a decreased GS was a poor prognostic factor in lenvatinib-treated u-HCC patients.

## 1. Introduction

The administration of tyrosine kinase inhibitors (TKIs) has become an important treatment option for improving the prognosis of patients with unresectable hepatocellular carcinoma (u-HCC). Lenvatinib, an oral multi-kinase inhibitor, is recommended as a first-line systemic chemotherapy for u-HCC, as well as sorafenib [1,2]. 

Sarcopenia is characterized by a loss of muscle strength associated with a progressive reduction of skeletal muscle mass [3]. Recently, sarcopenia has been recognized as a poor prognostic factor in various fields of clinical medicine, including cancer [4,5,6]. Sarcopenia has been reported to worsen the prognosis of patients, with u-HCC patients treated with sorafenib and lenvatinib [7,8,9,10]. Chronic liver disease (CLD), the underlying disease of HCC, is considered a representative cause of secondary sarcopenia, as the liver is a central organ of nutrient metabolism [3,11]. Due to the elevated energy expenditure and catabolism from the tumor burden, the underlying liver disease and adverse effects (AEs) of TKIs, u-HCC patients are considered to be at high risk for sarcopenia. The Eastern Cooperative Oncology Group performance status (ECOG-PS) has traditionally been used to assess the general activity; however, because the PS is a subjective evaluation, it has been pointed out that different evaluators may assign different values [12]. Thus, sarcopenia assessed using an objective approach is expected to complement the PS in the assessment of the general activity. Although the recent definition and diagnostic criteria for sarcopenia have emphasized muscle strength over muscle mass as the primary parameter of sarcopenia [13], most previous studies have only evaluated the effect of skeletal muscle mass loss on u-HCC patients treated with sorafenib or lenvatinib [7,8,9,10]. Indeed, no studies have examined the effects of muscle strength in u-HCC patients treated with TKIs.

This study evaluated the impact of sarcopenia-related factors (muscle strength and skeletal muscle mass) based on the established guidelines on the survival of u-HCC patients treated with lenvatinib.

## 2. Results

### 2.1. Baseline Characteristics 

Between April 2018 and April 2020, 69 patients with HCC received a lenvatinib treatment at our hospital. A total of six patients were excluded based on the defined exclusion criteria (lack of grip strength (*n* = 3), short observation period (*n* = 2) and lack of computed tomography (CT) findings (*n* = 1). Sixty-three patients were enrolled in this study. The clinical characteristics of the patients are shown in Table 1. The median age was 71 years, and 53 (84.1%) patients were men. Forty-four (69.8%) patients were Barcelona Clinic Liver Cancer classification (BCLC) C, and the modified albumin-bilirubin (mALBI) classifications were 1 (*n* = 18), 2a (*n* = 20) and 2b (*n* = 25). Forty-eight (76.2%) patients had recurrent HCC, and 12 (19.0%) and 4 (6.3%) patients had a history of sorafenib and regorafenib treatments, respectively. Twenty-two (34.9%) patients had low muscle mass (decreased skeletal muscle index (SMI)) and 21 (33.3%) had low muscle strength (decreased grip strength (GS)), respectively. Ultimately, 11 (17.5%) patients were diagnosed with sarcopenia. 

### 2.2. Results of the Lenvatinib Treatment in All Patients

The median observation period was 8.3 months. The estimated median survival time (MST) was 18.2 months (Figure 1A). The estimated median progression-free survival (PFS) was 6.0 months (Figure 1B). Of the 63 enrolled patients, 49 were diagnosed with disease progression. The median post-progression survival (PPS) was 7.1 months (Figure 1C). The responses of the patients were classified as follows: complete response (*n* = 0), partial response (*n* = 22), stable disease (*n* = 25), progressive disease (*n* = 15) and not evaluated (*n* = 1). The objective response rate (ORR) was 35.5%, and the disease control rate (DCR) was 75.8%. During the observation period, lenvatinib was discontinued in 30 patients due to PD (progressive disease), 16 due to AEs, 2 due to a decreased liver function, and 1 at the patient’s request; treatment was continued in all others.

### 2.3. Impact of mALBI on the OS, PFS and PPS

The OS of the mALBI 1/2a group was significantly higher than that of the mALBI 2b group (*p* < 0.01) (Figure 2A). The PFS of the mALBI 1/2a and ALBI 2b groups did not differ to a statistically significant extent (*p* = 0.48) (Figure 2B). The PPS of the mALBI 1/2a group was significantly longer than that of the mALBI 2b group (*p* < 0.01) (Figure 2C). 

### 2.4. The Comparison of the Normal and Decreased GS Groups and Normal and Decreased SMI Groups

The clinical characteristics of the patients with or without decreased GS or SMI are shown in Table 1. The decreased GS and SMI groups had significantly lower body mass indexes and body weights than the normal GS and SMI groups, respectively. The decreased GS group had a significantly higher maximum tumor diameter than the normal GS group. The mALBI-grade and Child-Pugh class in each group did not differ significantly. The respective ORR and DCR values were 35.0% and 75.0% in the decreased GS group, 35.7% and 76.2% in the normal GS group, 40.9% and 81.2% in the decreased SMI group and 32.5% and 72.5% in the normal SMI group; these differences were not statistically significant.

### 2.5. Influence of a Decreased GS and Decreased SMI on the OS, PFS and PPS

The OS of the normal GS group was significantly higher than that of the decreased GS group (*p* = 0.03) (Figure 3A). In contrast, the OS of the normal and decreased SMI groups did not differ significantly (*p* = 0.90) (Figure 3B).

There were no significant differences in the PFS between the normal GS and decreased GS groups or the normal SMI and decreased SMI groups (*p* = 0.24 and 0.44, respectively) (Figure 3C,D). The PPS for the normal GS groups were significantly higher in comparison to the decreased GS groups (*p* < 0.01) (Figure 3E). In contrast, the PPS of the normal SMI and decreased SMI groups did not differ significantly (*p* = 0.37) (Figure 3F).

### 2.6. AEs

Appendix A shows the AE profiles, which were observed in more than 15% of the overall study population. There was no significant difference in the time to discontinuation due to AEs between the normal SMI group and decreased SMI group or between the mALBI 1/2a group and mALBI 2b group (*p* = 0.64 and 0.24, respectively) (Figure 4A,C). The normal GS groups showed a significantly longer time to discontinuation due to AEs in comparison to the decreased GS groups (*p* < 0.01) (Figure 4B). 

### 2.7. Factors Associated with the OS

The results of the univariate and multivariate Cox proportional hazards model analyses are shown in Table 2. In the univariate analysis, three factors (mALBI 2b, α-fetoprotein (AFP) ≥ 400 ng/mL and decreased GS) had *p*-values of <0.1. The multivariate analysis revealed that the mALBI 2b (hazard ratio (HR) 4.39; 95% CI, 1.72–11.2; *p* < 0.01) and decreased GS (HR 3.55; 95% CI, 1.42–8.92) were independently associated with an increased risk of poor OS. The decreased SMI was not detected as a poor prognostic factor.

### 2.8. OS According to the Number of Risk Factors

We classified the enrolled patients into three groups according to the number of risk factors identified by the multivariate analysis (mALBI 2b and decreased GS). The patients with two risk factors had significantly lower OS in comparison to those with one or zero (*p* < 0.01) (Figure 5). 

### 2.9. Relationship between GS and SMI and the Hepatic Functional Reserve

To clarify the association between the muscle performance and hepatic reserve function, the GS and SMI were stratified by the mALBI grade (Figure 6). There was no significant difference in the mALBI and the SMI or GS. 

## 3. Discussion

To our knowledge, this is the first study to separately evaluate the impact of sarcopenia-related factors (muscle strength and mass) on the survival while complying with the established guidelines in u-HCC patients treated with lenvatinib. 

This study showed that a low muscle strength and low hepatic functional reserve were poor prognostic factors in u-HCC patients treated with lenvatinib. In addition, low muscle strength rather than low skeletal muscle mass was associated with a poor prognosis in u-HCC patients treated with lenvatinib. The present study therefore suggests that the skeletal muscle quality is more important than the quantity.

The results of the present study differ from those of previous retrospective studies that reported that low skeletal muscle mass was associated with poor tolerability and survival in HCC patients treated with sorafenib or lenvatinib [7,8,9,10]. There may be several reasons for this discrepancy. The method of measuring the skeletal muscle mass and the cut-off values differed among studies. In contrast, the measurement of the skeletal mass and strength in the present study complies with the established guideline (Japan Society of Hepatology (JSH) guideline) [14]. A previous cross-sectional study showed that skeletal muscle steatosis was significantly correlated with age [15]. In addition, our previous study reported that the intramuscular adipose tissue content increases 1.8%/year with aging [16]. Since the present study included a large number of older adults, it is possible that skeletal muscle mass does not strongly reflect the actual muscle function due to muscle steatosis—in elderly patients, in particular—which may have led to it being less likely to reflect the prognosis. In addition, the body weight is strongly correlated with the SMI [17], because lenvatinib, unlike sorafenib, has its dose adjusted by the body weight, which may have reduced the effect of the SMI. Indeed, the present study showed that the time to discontinuation due to AEs did not differ significantly between patients with and without a decreased SMI, although the patients in the decreased GS group discontinued treatment significantly earlier than those in the normal GS group. This result suggested that the SMI is adjusted to some extent by the body weight, while the GS is not adjusted, which may have led to an earlier discontinuation of lenvatinib and a consequent poor prognosis. Furthermore, the muscle strength was lost two-to-five times faster than the muscle mass, indicating that muscle strength is a more sensitive indicator of prognosis than skeletal muscle mass [18]. Although our cohort had a preserved hepatic reserve at the start of lenvatinib, 19% of patients had a history of TKIs, and 76.2% patients had recurrent HCC. Since HCC often recurs and requires repeated treatments, the loss of muscle strength may have been a stronger factor than the loss of skeletal muscle mass in the process. Indeed, recently published studies have reported that GS (muscle strength) has a greater effect on the prognosis than the skeletal muscle mass in community-dwelling elderly people, as well as patients with chronic liver disease [19,20,21]. In fact, the European Working Group on Sarcopenia in Older People (EWGSOP) guidelines state that low muscle strength is the primary parameter of sarcopenia, because muscle strength is the most reliable measure of the muscle function [13]. The EWGSOP guidelines recommend the use of GS as an indicator of muscle strength, because the GS is a simple and inexpensive method and is a powerful predictor of poor patient outcomes (e.g., longer hospital stays, increased functional limitations, poor health-related quality of life and death). In addition, the measurement of GS is usually easier than the measurement of skeletal muscle mass, because the measurement of skeletal muscle mass requires special equipment, such as a bioelectrical impedance analysis (BIA), dual-energy X-ray absorptiometry (DEXA) and CT. Thus, the GS is useful as a simple, objective prognostic indicator in clinical practice that can be measured before treatment.

There was no significant difference in the ORR, DCR or PFS between the groups with and without decreased GS; however, the differences in PPS may have contributed to the differences in OS. This might be because normal GS patients can still receive other treatments, including TKIs, after disease progression, while decreased GS patients cannot be treated due to the deterioration of their PS or liver function. A lack of difference in treatment responses to lenvatinib between those with and without a decreased GS may have been due to the relatively high number of patients who received a reduced dose at the initiation of the lenvatinib treatment in this study.

In the present study, as in previous reports, patients with an mALBI grade of 1/2a were considered the best candidates for the lenvatinib treatment [22,23,24]. However, lenvatinib may have to be initiated for patients with an mALBI-grade 2b in clinical practice. Our results suggested that, in patients without a decreased GS, lenvatinib may also be selected, even for patients with an mALBI grade of 2b. However, the lenvatinib treatment for mALBI-grade 2b patients with a decreased GS should be performed with care or avoided. However, the number of cases is small, and further research is needed. It is well-known that the prognosis of HCC patients is dependent on the tumor burden and hepatic reserve function [25]. In addition to these factors, the present study demonstrated that muscle strength was an important prognostic factor.

Recently, the prognostic impact of changes in body composition caused by sorafenib has been reported. Previous retrospective studies reported that the rapid depletion of skeletal muscle mass or subcutaneous adipose tissue was a poor prognostic factor for u-HCC patients treated with sorafenib [26,27]. In the present study, we evaluated the skeletal muscle condition only at the start of lenvatinib, so a further study is needed to confirm the prognostic impact of the changes in the body composition, including adipose tissue and muscle, in u-HCC patients treated with lenvatinib.

Although the liver is theoretically closely related to skeletal muscle, given its involvement in protein synthesis and ammonia metabolism, the present study showed no consistent relationship between the hepatic functional reserve and skeletal muscle strength or mass. This may have been influenced by the small sample size, history of HCC treatment and comorbidities.

While the present study examined the impact of sarcopenia-related factors on lenvatinib, whether or not same results would be obtained with other types of systemic therapy remains unclear. However, two recent studies of systemic chemotherapy for patients with advanced cancer reported that a low grip strength was a poor prognostic factor [28,29]. These studies included a variety of tumors, including lung, colorectal, breast and prostate cancers, suggesting that the effect of GS may not be independent of the cancer type. Furthermore, it has already been shown that GS has a stronger prognostic impact than SMI in CLD, the underlying disease of HCC [19,21]. These results suggest that a low GS is a poor prognostic factor for systemic chemotherapy other than lenvatinib in u-HCC. Further studies are therefore needed to confirm these findings for other regimens in u-HCC.

The present study was associated with some limitations. First, it was a single-center study with a relatively small sample size and a short observation period. Second, this study did not measure the GS and SMI repeatedly, so the changes in these values during the treatment are unknown. Third, our cohort only included Asian subjects. Since the cut-off value of skeletal muscle mass and GS vary depending on race/ethnicity, further studies are needed to confirm these findings in other regions.

## 4. Materials and Methods

### 4.1. Patients

This single-center cohort study was based on data collected from a university hospital. We analyzed patients with u-HCC who were treated with lenvatinib (Lenvima^®^, Eisai Co., Ltd., Tokyo, Japan) in our hospital between April 2018 and April 2020. The diagnosis of HCC was confirmed according to the European Association for the Study of the Liver Clinical Practice guidelines [30]. U-HCC was confirmed by pathological or radiographic findings. The study complied with the provisions of the Declaration of Helsinki and was approved by the Ethics Committee of our hospital (MH2019-082).

### 4.2. Lenvatinib Therapy

Lenvatinib therapy was indicated for patients with Child-Pugh A or B in u-HCC and an ECOG-PS of 0 or 1 and/or (a) extrahepatic metastasis, (b) the presence of vascular invasion and (c) refractory to previous transcatheter arterial chemoembolization (TACE). Patients for whom the observation period was <28 days were excluded. All patients were admitted to the hospital for one week for lenvatinib induction to check for AEs. They also received nutritional guidance according to the European Society for Clinical Nutrition and Metabolism guidelines [31].

In principle, patients received lenvatinib (12 mg, once-daily for those with body weights (BW) ≥ 60 kg at baseline and 8 mg, once-daily for those with BW < 60 kg at baseline). For patients with a risk factor, such as Child-Pugh class B or comorbidities, the initial dose of lenvatinib could be reduced from 12 mg to 8 mg or 4 mg and from 8 mg to 4 mg. During treatment, clinicians could adjust the daily dose of lenvatinib according to the severity of the AEs. According to the guideline provided by the manufacturer, the drug dose was reduced or treatment was interrupted in patients who developed grade ≥ 3 severe AEs or any unacceptable grade 2 drug-related AEs. AEs were assessed according to the National Cancer Institute Common Terminology Criteria for Adverse Events, version 4.0. Lenvatinib was continued until disease progression was diagnosed or uncontrollable AEs occurred or if the patient decided that they did not wish to continue treatment. When the HCC progression was observed after the initial therapy, the most appropriate therapy was performed according to the clinical guidelines [32].

### 4.3. Assessment and Follow-up

The therapeutic response was evaluated once every 6–8 weeks, according to the modified Response Evaluation Criteria in Solid Tumors (mRECIST), using dynamic CT or magnetic resonance imaging (MRI) [33]. The hepatic reserve function was assessed using the Child-Pugh classification [34] and mALBI, as previously reported [35]. Patients with mALBI 1/2a were compared with those with mALBI 2b. The HCC stage was determined according to the Barcelona Clinic Liver Cancer classification [30].

### 4.4. The Diagnosis and Cut-off Value of Sarcopenia-Related Factors

As in our previous reports, the SMI was calculated by dividing the skeletal muscles mass (cm^2^) by the square of the height (cm^2^/m^2^) using abdominal CT performed within one month of the initiation of Lenvatinib [16]. GS was measured as an indicator of muscle strength, using a Smedley-type digital hand dynamometer (T.K.K.5401; Takei Scientific Instruments, Niigata, Japan) with the elbow straight in the standing position. The maximal strength values of two trials for each hand were averaged for the analysis. GS was measured on the day of lenvatinib initiation. The cut-off values of the sarcopenia-related factors were based on the Japan Society of Hepatology guidelines for sarcopenia in liver disease [14]. Low muscle strength was defined as a GS of <26 kg and <18 kg in men and women, respectively. Low muscle volume was defined as an SMI < 42 cm^2^/m^2^ and <38 cm^2^/m^2^ in men and women, respectively.

### 4.5. Statistical Analysis

Continuous variables are expressed as the median and range. Categorical variables are expressed in numbers and percentages (%). We used the Mann-Whitney U test to analyze continuous variables and Fisher’s exact test to analyze categorical variables. Overall survival (OS; time from lenvatinib treatment to any cause of death), progression-free survival (PFS; time from lenvatinib treatment to progression or any cause of death), post-progression survival (PPS; time from disease progression after lenvatinib treatment to death) and time to discontinuation due to AEs (time from the lenvatinib treatment to discontinuation due to AEs) were analyzed using the Kaplan-Meier method, and the difference between the two groups was compared using the log-rank test. Factors potentially associated with poor OS were assessed using univariate and multivariate Cox’s proportional hazards regression models. Factors with *p*-values of <0.1 in a univariate analysis were included in the multivariate analysis. The following variables were included in a univariate analysis: age, sex, mALBI grade, Child-Pugh class, α-fetoprotein (AFP), des-γ-carboxy prothrombin (DCP), BCLC stage, reduced initial dose of lenvatinib, maximum tumor diameter, tumor number, vascular invasion, extrahepatic metastasis, decreased GS and decreased SMI. All tests were two-sided. *P*-values of <0.05 were considered to indicate statistical significance. All statistical analyses were performed using EZR (Saitama Medical Center, Jichi Medical University, Saitama, Japan), which is a graphical user interface for the R software program (version 3.3.2, the R Foundation for statistical computing, Vienna, Austria) [36].

## 5. Conclusions

In addition to a low hepatic functional reserve, low muscle strength was a poor prognostic factor in u-HCC patients treated with lenvatinib.

## Figures and Tables

**Figure 1 cancers-12-02146-f001:**
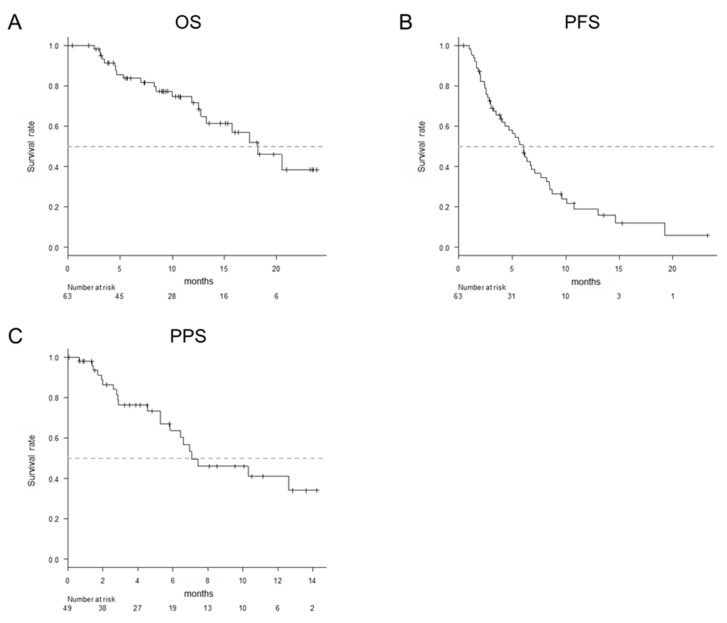
The overall survival (OS), progression-free survival (PFS) and post-progression survival (PPS) in all patients. The median survival time (MST) was 18.2 months (**A**). The median PFS was 6.0 months (**B**). The median PPS was 7.1 months (**C**).

**Figure 2 cancers-12-02146-f002:**
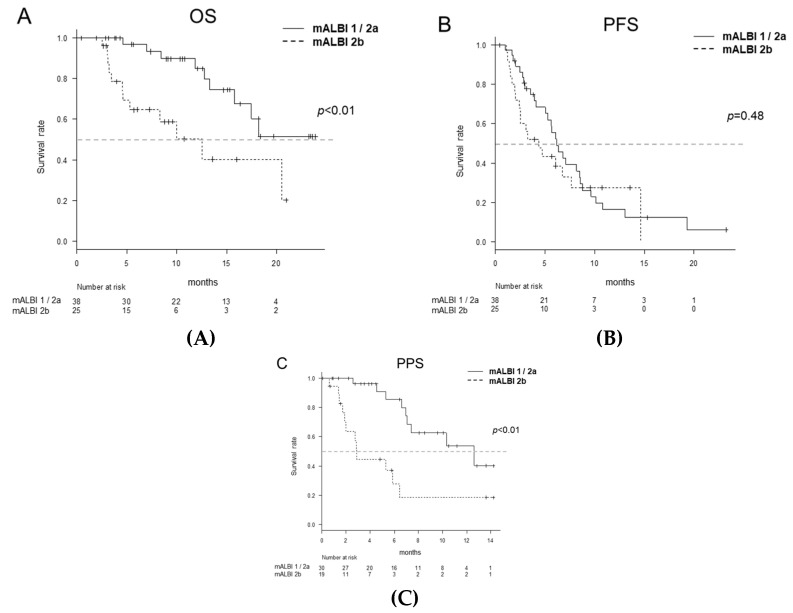
The overall survival (OS), progression-free survival (PFS) and post-progression survival (PPS) according to the modified albumin-bilirubin (mALBI). The median survival time (MST) was not reached in the mALBI 1/2a group and 12.5 months in the mALBI 2b group (*p* < 0.01) (**A**). The median PFS was 6.2 months in the mALBI 1/2a group and 4.4 months in the mALBI 2b group (*p* = 0.48) (**B**). The median PPS was 12.6 months in the mALBI 1/2a group and 2.9 months in the mALBI 2b group (*p* < 0.01) (**C**).

**Figure 3 cancers-12-02146-f003:**
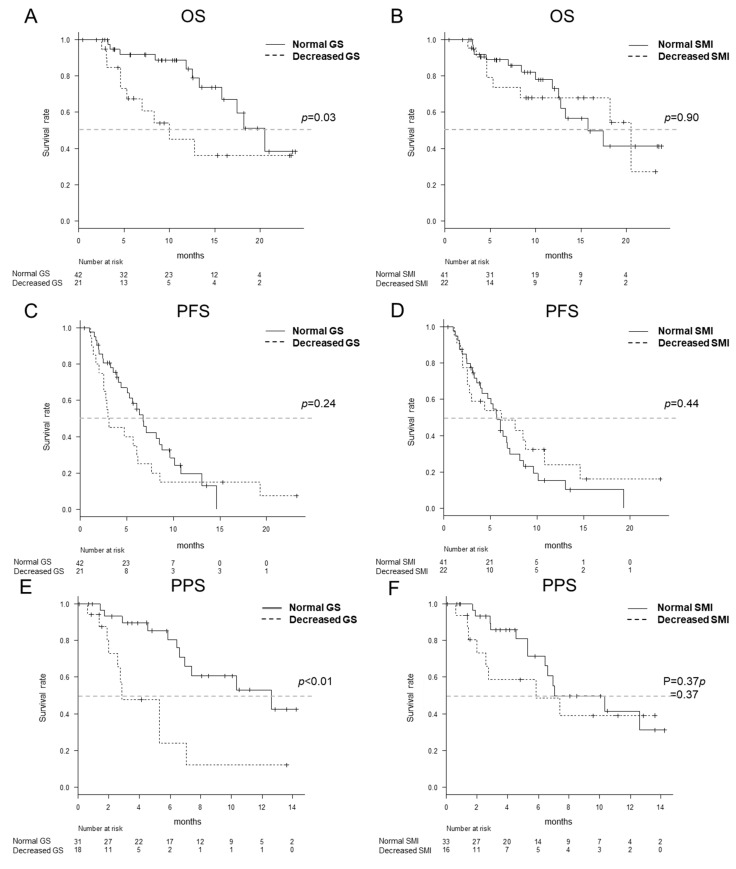
The overall survival (OS), progression-free survival (PFS) and post-progression survival (PPS) according to the grip strength (GS) and skeletal muscle index (SMI). The MST was 20.5 months in the normal GS group and 10.0 months in the decreased GS group (*p* = 0.03) (**A**) and 15.8 months in the normal SMI group and 20.5 months in the decreased SMI group, respectively (*p* = 0.90) (**B**). The median PFS was 6.7 months in the normal GS group and 3.0 months in the decreased GS group (*p* = 0.24) (**C**) and 5.7 months in the normal SMI group and 6.2 months in the decreased SMI group, respectively (*p* = 0.44) (**D**). The median PPS was 12.6 months in the normal GS group and 2.9 months in the decreased GS group (*p* < 0.01) (**E**) and 7.1 months in the normal SMI group and 5.8 months in the decreased SMI group, respectively (*p* = 0.37) (**F**).

**Figure 4 cancers-12-02146-f004:**
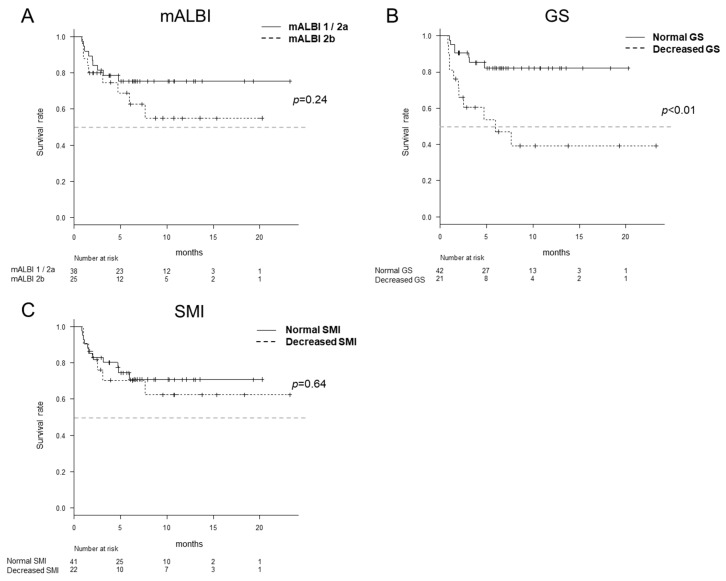
Discontinuation due to adverse effects (AEs) according to the mALBI, grip strength (GS) and skeletal muscle index (SMI). There was no significant difference in the time to discontinuation due to AEs between the mALBI 1/2a group and mALBI 2b group or between the normal SMI group and decreased SMI group (*p* = 0.24 and 0.64, respectively) (**A**,**C**). The time to discontinuation due to AEs for the normal GS group was significantly longer in comparison to the decreased GS group (*p* < 0.01) (**B**).

**Figure 5 cancers-12-02146-f005:**
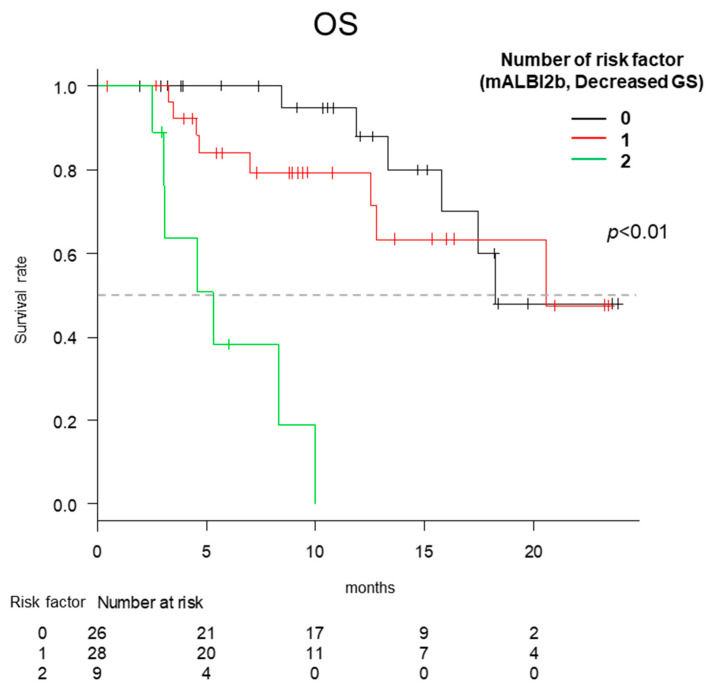
The overall survival (OS) according to the number of risk factors. The patients with 2 risk factors (modified albumin-bilirubin-grade (mALBI) 2b and decreased grip strength (GS)) had a significantly lower OS than those with 1 or 0 (*p* < 0.01).

**Figure 6 cancers-12-02146-f006:**
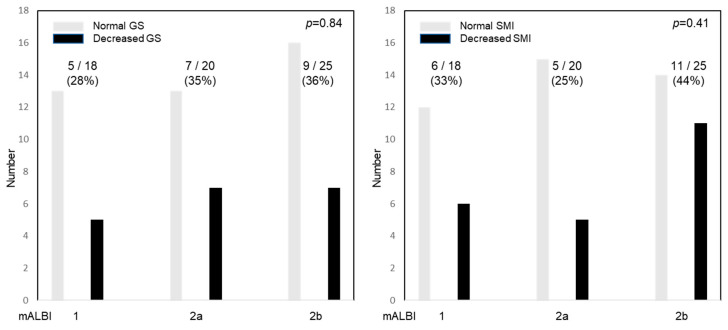
Relationship between the grip strength (GS) and skeletal muscle index (SMI) and mALBI. The numbers represent the percentage of patients with decreased GS and SMI. There was no significant difference between the mALBI and GS (*p* = 0.84) or SMI (*p* = 0.41).

**Table 1 cancers-12-02146-t001:** The baseline characteristics and comparison of the normal GS and decreased GS or normal SMI and decreased SMI groups.

	All Patients (*n* = 63)	Normal GS Group (*n* = 42)	Decreased GS Group (*n* = 21)	*p*	Normal SMI Group (*n* = 41)	Decreased SMI Group (*n* = 22)	*p*
Sarcopenia, yes/no	11/52						
Gender, male/female	53/10	38/4	15/6	0.07	37/4	16/6	0.08
Age (years)	71 (50–86)	70 (50–81)	74 (61–86)	0.06	70 (50–84)	72 (57–86)	0.36
Body mass index (kg/m^2^)	23.2 (16.4–32.2)	24.2 (17.0–32.2)	22.5 (16.4–28.7)	0.04	24.3 (18.6–32.2)	20.4 (16.4–31.5)	<0.01
Body weight (kg)	61.9 (35.4–102.1)	64.4 (45.2–102.1)	55.2 (35.4–75.5)	<0.01	65.0 (45.6–102.1)	53.8 (35.4–86.8)	<0.01
Etiology, HBV/HCV/Alcohol/others	10/23/17/13	7/19/11/5	3/4/6/8	0.07	7/17/11/6	3/6/6/7	0.44
mALBI. 1/2a/2b	18/20/25	13/13/16	5/7/9	0.89	12/15/14	6/5/11	0.43
Child-Pugh class, A/B	49/14	32/10	17/4	0.76	34/7	15/7	0.21
ECOG-PS	57/6	40/2	17/4	0.09	38/3	19/3	0.41
Albumin (g/dL)	3.6 (2.6–4.6)	3.7 (2.8–4.6)	3.4 (2.6–4.2)	0.08	3.7 (2.6–4.6)	3.4 (2.8–4.3)	0.21
AST (IU/L)	44 (19–154)	46 (21–133)	41 (19–154)	0.40	44 (19–133)	51 (19–154)	0.57
ALT (IU/L)	34 (8–123)	41 (15–123)	24 (8–106)	0.02	40 (14–123)	29 (8–106)	0.12
Total bilirubin (IU/L)	0.7 (0.2–1.6)	0.8 (0.3–1.6)	0.6 (0.2–1.1)	0.013	0.7 (0.2–1.6)	0.7 (0.2–1.1)	0.28
Creatinine (mg/dl)	0.80 (0.49–2.18)	0.79 (0.49–1.62)	0.83 (0.52–2.18)	0.62	0.80 (0.49–2.18)	0.81 (0.52–1.57)	0.67
Prothrombin (INR)	1.09 (0.90–1.49)	1.06 (0.90–1.49)	1.14 (0.99–1.29)	0.17	1.08 (0.90–1.49)	1.13 (0.99–1.28)	0.67
Platelets (×10^4^/μL)	14.1 (3.0–59.8)	12.3 (3.0–58.7)	22.1 (7.7–59.8)	<0.01	13.6 (3.0–58.7)	14.4 (4.7–59.8)	0.74
AFP <400/>400 (ng/mL)	46/17	30/12	16/5	0.77	31/10	15/7	0.56
DCP <400/>400 (mAU/mL)	29/34	17/25	12/9	0.29	18/23	11/11	0.79
Maximum tumor size (cm)	5.8 (1.6–22.2)	5.0 (1.6–13.0)	8.5 (2.0–22.2)	<0.01	5.0 (1.6–22.2)	7.3 (2.0–17.4)	0.052
Tumor number (single/multiple)	4/59	2/40	2/19	0.60	2/39	2/20	0.61
Vascular invasion, yes/no	36/27	21/21	15/6	0.18	23/18	13/9	0.99
Extrahepatic metastasis, yes/no	23/40	13/29	10/11	0.27	14/27	9/13	0.60
BCLC classification (B/C)	19/44	16/26	3/18	0.08	15/26	4/18	0.15
Reduced dose at initial lenvatinib, yes/no	24/39	12/30	12/9	0.053	14/27	10/12	0.42
HCC, naïve/recurrence	15/48	10/32	5/16	1.00	10/31	5/17	1.00
Sorafenib naïve/experience	51/12	35/7	16/5	0.51	34/7	17/5	0.74
Regorafenib naïve/experience	59/4	40/2	19/2	0.60	38/3	21/1	1.00
Objective response rate (%)	35.5	35.7	35.0	1.00	32.5	40.9	0.80
Disease control rate (%)	75.8	76.2	75.0	1.00	72.5	81.2	0.84

GS, grip strength; SMI, skeletal muscle index; HBV, hepatitis B virus; HCV, hepatitis C virus; ECOG-PS. Eastern Cooperative Oncology Group performance status; mALBI, modified albumin-bilirubin; AST, aspartate aminotransferase; ALT, alanine aminotransferase; INR, international normalized ratio; AFP, α-fetoprotein; DCP, des-γ-carboxy prothrombin; BCLC, Barcelona Clinic Liver Cancer and HCC, hepatocellular carcinoma.

**Table 2 cancers-12-02146-t002:** Factors associated with poor survival.

Variables	Univariate Analysis			Multivariate Analysis		
	HR	95% CI	*p*	HR	95% CI	*p*
Gender, female	2.07	0.75–5.69	0.15			
Age (y)	1.04	0.98–1.10	0.18			
mALBI 2b (vs. 1/2a)	3.32	1.38–7.97	<0.01	4.39	1.72–11.2	<0.01
Child-Pugh class B	2.14	0.82–5.58	0.12			
AFP > 400 (ng/mL)	2.14	0.87–5.29	0.09			
DCP > 400 (mAU/mL)	2.18	0.79–6.01	0.13			
BCLC C	0.87	0.34–2.19	0.77			
Maximum tumor size (cm)	1.04	0.95–1.14	0.39			
Tumor number, multiple	1.01	0.13–7.65	0.99			
Vascular invasion, yes	0.98	0.42–2.35	0.97			
Extrahepatic metastasis, yes	1.04	0.42–2.54	0.93			
Reduced initial dose of Lenvatinib, yes	1.85	0.75–4.54	0.18			
Decreased grip strength, yes	2.57	1.08–6.09	0.03	3.55	1.42–8.92	<0.01
Decreased SMI, yes	1.06	0.43–2.56	0.90			

Cox proportional hazard model. mALBI, modified albumin-bilirubin; AFP, α-fetoprotein; DCP, des-γ-carboxy prothrombin; BCLC, Barcelona Clinic Liver Cancer; GS, grip strength and SMI, skeletal muscle index.

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
