# Peer review of "Impact of Grip Strength in Patients with Unresectable Hepatocellular Carcinoma Treated with Lenvatinib"

_cancers, 2020, doi:10.3390/cancers12082146_

Round 1

Reviewer 1 Report

All abbreviations must be explained at the first appearance in the text. In the same way all abbreviations must be explicit in the caption of tables and figures. Please check the text.

Although editorially it is possible to do otherwise I would prefer to have the discussion after the materials and methods, just before the conclusions.

The paper is interesting because it takes into consideration some potential indicator of clinical response and prognosis regarding a recently labeled treatment. I would recommend to report and anlyze also the global performance status assessed using ECOG method, which is also part of the BCLC classification. 

I also suggest to try to stratify GS and SMI values according the Child score to somehow describe the weight of the underlying liver disease in the observed muscle performance

Reviewer 2 Report

Kei Endo and collaborators report the impact of sarcopenia indicators such as decrease of the grip strength (GS) and a lower skeletal muscle index (SMI) on the survival of patients with unresectable hepatocellular carcinoma (u-HCC) treated with Lenvatinib.

Contrary to the previous reported works in the field, which have associated SMI with survival and tolerability to lenvatinib in treated HCC patients, this study did not show an association of SMI with overall or progression-free survival, or with the discontinuation due to AEs.

Instead, their results point towards the implication of the GS in the overall and post-progression survival, together with modified albumin-bilirubin (mALBI) grade. GS was also the only variable associated with discontinuation due to AEs.

In addition, they define which of the variables are independent predictors of mortality using multivariate Cox proportional hazards models, and use a combination of variables to predict overall survival.

In my opinion, this is a work that is useful for the selection of possible sarcopenia-related markers associated with different aspects of the treatment of HCC with Lenvatinib, for which not many studies are yet accomplished. Considering the minor comments that I detail bellow, I would elaborate more in both the introduction and the discussion on the impact in the field of discriminating between the muscle function and muscle mass, and why this is relevant for the management of these novel treatments indicated in specific cases of HCC.

Minor comments:

-  Why treatment failure is not assessed?

- Discuss why considering the GS in addition to SMI (irrespective of this being also associated with better outcomes) would be advantageous in light of your results.

- Would you consider the nutritional state as a variable?

- Maybe you could discuss the interaction between a low hepatic functional reserve and the co-occurrence of sarcopenia. This is also interesting given the combination of mALBI grade and GS in the prediction of overall survival.

- In the introduction, the authors should elaborate more on the justification of the study, the link of sarcopenia with TKi and cancer in general and in the stages/conditions of HCC where TKi are indicated, on the current markers of survival related to sarcopenia and why more informative markers are still needed. Also, on the recent progress of Lenvatinib treatment for HCC in the context of the search of markers of outcome for this therapy.

- In figures 1 to 4, the line that is labelled in blue is not blue in the curve.

- In figure 2, I understand that the median MST, 18.2, was not reached in both mALBI 1/2a and mALBI 2b groups.

- In the discussion they should mention the extended impact this work has on the management of treatment with other TKi, on the information about the interaction sarcopenia or even fat loss with Sorafenib, which is much more characterized. In line 152, I would substitute “quantity” by “mass”. In line 163, I would specify the measure of skeletal mass and strength instead of the diagnosis of sarcopenia, since all the work is centered specifically in these two sarcopenia parameters rather than in the diagnosis of sarcopenia. Line 198, “the” should be removed. In line 243, more elaboration on the utility of muscle strength to be included as a prognostic factor would be interesting.

- Also in the discussion, when they suggest the intramuscular adipose content in the elderly to support the different results with respect the other studies, they should comment on the distributions per age and maybe skeletal muscle steatosis in those reports.

Reviewer 3 Report

This article described the prognostic factor in lenvatinib-treated u-HCC.

Author reported multivariate Cox proportional hazards model showed that ALBI2b and a decreased grip strength were independently associated with an increased risk of poor prognosis.

Many readers show a deep interest in TKI treatment for u-HCC, especially experts in this field, are greatly interested in this issue. This article has been written well. However, a minor concern arose as mentioned below.

Minor Point

1. The authors should add the number of patients with sarcopenia in Table 1.

2. Imai K et al. reported rapid depletion of the subcutaneous fat mass index (SFMI) and skeletal muscle index (SMI) after the introduction of sorafenib indicates a poor prognosis (Imai K et al. Cancers 2019, 11, 1206).

Since the images have been evaluated by CT, it is considered possible to change the SMI.

In this study, SMI was not demonstrated as a prognostic factor in lenvatinib, but how about the change in SMI (ΔSMI)?

If the author has data on the rate of change in grip strength, the authors should evaluate and discuss the prognostic impact of ΔSMI or Δgrip strength (GS).

3. The post-progression survival was significantly lower in the GS-decreased group than in the GS-normal group, and the authors attributed this to decreased performance status (PS) or hepatic functional reserve. The authors should describe clearly the difference in PS and hepatic reserve between both groups.

4. Although sarcopenia has attracted in clinical in recent years, PS has been the traditional method of assessing a patient's general condition. On the other hand, because PS is a subjective method, it has been pointed out that different evaluators may have different outcomes (E Neeman, et al. Oncologist. 2019;24:e1460-6). If authors have the PS data, authors should evaluate the impact of PS.

Reviewer 4 Report

In this manuscript, authors evaluated the impact of sarcopenia-related factors such as grip strength (GS) and skeletal muscle mass based on established guidelines on the survival of 63 u-HCC patients treated with lenvatinib. GS amd SMI was defined according to the Japan Society of Hepatology (JSH) guidelines for sarcopenia using a liver disease algorithm.

According to author’s statement, this is the first analysis for the impact of GS and SMI based on the JSH guidelines on OS, PFS, PPS. Baseline ALBI1/2a vs ALBI2b status, normal GS vs decreased GS, were associated with OS, but PFS was not different between normal GS vs decreased GS. Discontinuation due to AEs were associated with normal GS vs decreased GS. Based on multivariate analyses based on OS, authors concluded that in addition to a low hepatic functional reserve, decreased GS in particular was a poor prognostic factor for OS in u-HCC patients treated with lenvatinib. In addition, patient with both a low hepatic function and decreased GS had a poor OS comparer to patients with either a low hepatic function or decreased GS, or without them.

  • Association of decreased GS with OS is clear, and it is an independent factor prognostics together with liver function after multivariate analysis, and especially GS measurements are easy to be done by individual patients. 
  • It is interesting and informative that ALBI2b status with normal GS have a better OS. It is may be informative to treat those patients with lenvatinib, but it not clear if short OS of patients with ALBI2b status and decreased GS will occurre with other therapy.
  • One of difficulty in this analysis was no association of PFS with decreased GS measurements in this study, and all patient were treated only with lenvatinib to discuss prognostic roles of decreased GS in this indication.
  • Authors showed clear associations of decreased GS with time to discontinuation demonstrating that decreased GS was associated with earlier discontinuation of Lenvatinib. As authors discussed, dose reductions of lenvatinib might occurred, and those adjustments might affect PFS despite of GS at baseline, and then no association with PFS. Monitoring of changes of GS may be more informative to be improvements of PFS and OS.
  • Association of decreased GS with OS, but not with PFS, in patients with lenvatinib treatment is clear, but this results need to be tested for other SOC drugs in this indication to determine if this observation is common prognostics factors for any SOC drugs, or selective for lenvatinib therapy.

Author Response

RESPONSE TO REVIEWER 4

In this manuscript, authors evaluated the impact of sarcopenia-related factors such as grip strength (GS) and skeletal muscle mass based on established guidelines on the survival of 63 u-HCC patients treated with lenvatinib. GS amd SMI was defined according to the Japan Society of Hepatology (JSH) guidelines for sarcopenia using a liver disease algorithm.

According to author’s statement, this is the first analysis for the impact of GS and SMI based on the JSH guidelines on OS, PFS, PPS. Baseline ALBI1/2a vs ALBI2b status, normal GS vs decreased GS, were associated with OS, but PFS was not different between normal GS vs decreased GS. Discontinuation due to AEs were associated with normal GS vs decreased GS. Based on multivariate analyses based on OS, authors concluded that in addition to a low hepatic functional reserve, decreased GS in particular was a poor prognostic factor for OS in u-HCC patients treated with lenvatinib. In addition, patient with both a low hepatic function and decreased GS had a poor OS comparer to patients with either a low hepatic function or decreased GS, or without them.

Association of decreased GS with OS is clear, and it is an independent factor prognostics together with liver function after multivariate analysis, and especially GS measurements are easy to be done by individual patients. 

It is interesting and informative that ALBI2b status with normal GS have a better OS. It is may be informative to treat those patients with lenvatinib, but it not clear if short OS of patients with ALBI2b status and decreased GS will occurre with other therapy.

One of difficulty in this analysis was no association of PFS with decreased GS measurements in this study, and all patient were treated only with lenvatinib to discuss prognostic roles of decreased GS in this indication.

Authors showed clear associations of decreased GS with time to discontinuation demonstrating that decreased GS was associated with earlier discontinuation of Lenvatinib. As authors discussed, dose reductions of lenvatinib might occurred, and those adjustments might affect PFS despite of GS at baseline, and then no association with PFS. Monitoring of changes of GS may be more informative to be improvements of PFS and OS.

Association of decreased GS with OS, but not with PFS, in patients with lenvatinib treatment is clear, but this results need to be tested for other SOC drugs in this indication to determine if this observation is common prognostics factors for any SOC drugs, or selective for lenvatinib therapy.

Response: While the present study examined the impact of sarcopenia-related factors on lenvatinib, whether or not same results would be obtained with other TKIs is unclear. This has now been mentioned in the limitations, as follows:

Line 254-255

“Fourth, the present study revealed the impact of sarcopenia-related factors in lenvatinib, however, whether or not the same results would be obtained with other TKIs is unclear.”

Round 2

Reviewer 4 Report

In this manuscript, authors evaluated the impact of sarcopenia-related factors such as grip strength (GS) and skeletal muscle mass based on established guidelines on the survival of 63 u-HCC patients treated with lenvatinib. GS amd SMI was defined according to the Japan Society of Hepatology (JSH) guidelines for sarcopenia using a liver disease algorithm.

Based on multivariate analyses based on OS, authors concluded that in addition to a low hepatic functional reserve, decreased GS in particular was a poor prognostic factor for OS in u-HCC patients treated with lenvatinib. In addition, patient with both a low hepatic function and decreased GS had a poor OS comparer to patients with either a low hepatic function or decreased GS, or without them.

It is interesting and informative that ALBI2b status with normal GS have a better OS. It is may be informative to treat those patients with lenvatinib, but it not clear if short OS of patients with ALBI2b status and decreased GS will occurre with other therapy.

One of difficulty in this analysis was no association of PFS with decreased GS measurements in this study, and all patient were treated only with lenvatinib to discuss prognostic roles of decreased GS in this indication.

Authors showed clear associations of decreased GS with time to discontinuation demonstrating that decreased GS was associated with earlier discontinuation of Lenvatinib. As authors discussed, dose reductions of lenvatinib might occurred, and those adjustments might affect PFS despite of GS at baseline, and then no association with PFS. Monitoring of changes of GS may be more informative to be improvements of PFS and OS.

Association of decreased GS with OS, but not with PFS, in patients with lenvatinib treatment is clear, but this results need to be tested for other SOC drugs in this indication to determine if this observation is common prognostics factors for any SOC drugs, or selective for lenvatinib therapy.

Authors discussed the limitation of this study in discussion section, as following

While the present study examined the impact of sarcopenia-related factors on lenvatinib, whether or not same results would be obtained with other TKIs is unclear.

Therefore, a further analysis of impact of GS in uHCC treated with other SOC drugs in uHCC also needs to be discussed.
